# LEARNING PARTICLE DYNAMICS FOR MANIPULATING RIGID BODIES, DEFORMABLE OBJECTS, AND FLUIDS

**Yunzhu Li, Jiajun Wu, Russ Tedrake, Joshua B. Tenenbaum, & Antonio Torralba**
Computer Science and Artificial Intelligence Laboratory
Massachusetts Institute of Technology
`{liyunzhu,jiajunwu,russt,jbt,torralba}@mit.edu`

## ABSTRACT

Real-life control tasks involve matters of various substances—rigid or soft bodies, liquid, gas—each with distinct physical behaviors. This poses challenges to traditional rigid-body physics engines. Particle-based simulators have been developed to model the dynamics of these complex scenes; however, relying on approximation techniques, their simulation often deviates from real-world physics, especially in the long term. In this paper, we propose to learn a particle-based simulator for complex control tasks. Combining learning with particle-based systems brings in two major benefits: first, the learned simulator, just like other particle-based systems, acts widely on objects of different materials; second, the particle-based representation poses strong inductive bias for learning: particles of the same type have the same dynamics within. This enables the model to quickly adapt to new environments of unknown dynamics within a few observations. We demonstrate robots achieving complex manipulation tasks using the learned simulator, such as manipulating fluids and deformable foam, with experiments both in simulation and in the real world. Our study helps lay the foundation for robot learning of dynamic scenes with particle-based representations.

## 1 INTRODUCTION

Objects have distinct dynamics. Under the same push, a rigid box will slide, modeling clay will deform, and a cup full of water will fall with water spilling out. The diverse behavior of different objects poses challenges to traditional rigid-body simulators used in robotics (Todorov et al., 2012; Tedrake & the Drake Development Team, 2019). Particle-based simulators aim to model the dynamics of these complex scenes (Macklin et al., 2014); however, relying on approximation techniques for the sake of perceptual realism, their simulation often deviates from real world physics, especially in the long term. Developing generalizable and accurate forward dynamics models is of critical importance for robot manipulation of distinct real-life objects.

We propose to learn a differentiable, particle-based simulator for complex control tasks, drawing inspiration from recent development in differentiable physical engines (Battaglia et al., 2016; Chang et al., 2017). In robotics, the use of differentiable simulators, together with continuous and symbolic optimization algorithms, has enabled planning for increasingly complex whole body motions with multi-contact and multi-object interactions (Toussaint et al., 2018). Yet these approaches have only tackled local interactions of rigid bodies. We develop dynamic particle interaction networks (DPI-Nets) for learning particle dynamics, focusing on capturing the dynamic, hierarchical, and long-range interactions of particles (Figure 1a-c). DPI-Nets can then be combined with classic perception and gradient-based control algorithms for robot manipulation of deformable objects (Figure 1d).

Learning a particle-based simulator brings in two major benefits. First, the learned simulator, just like other particle-based systems, acts widely on objects of different states. DPI-Nets have successfully captured the complex behaviors of deformable objects, fluids, and rigid-bodies. With learned DPI-Nets, our robots have achieved success in manipulation tasks that involve deformable objects of complex physical properties, such as molding plasticine to a target shape.

---

Our project page: `http://dpi.csail.mit.edu`

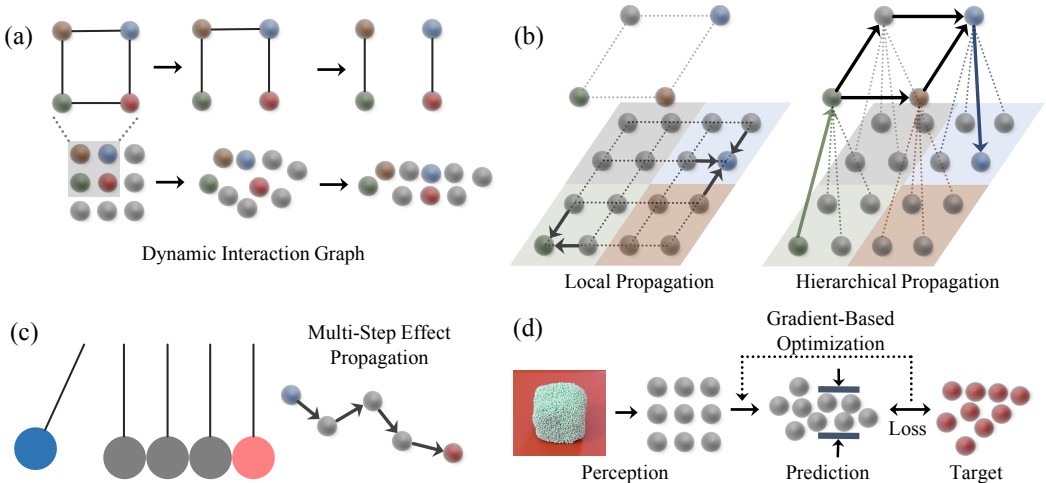

Figure 1: **Learning particle dynamics for control.** (a) DPI-Nets learn particle interaction while dynamically building the interaction graph over time. (b) Build hierarchical graph for multi-scale effect propagation. (c) Multi-step message passing for handling instantaneous force propagation. (d) Perception and control with the learned model. Our system first reconstructs the particle-based shape from visual observation. It then uses gradient-based trajectory optimization to search for the actions that produce the most desired output.

Second, the particle-based representation poses strong inductive bias for learning: particles of the same type have the same dynamics within. This enables the model to quickly adapt to new environments of unknown dynamics within a few observations. Experiments suggest that DPI-Nets quickly learn to adapt to characterize a novel object of unknown physical parameters by doing online system identification. The adapted model also helps the robot to successfully manipulate object in the real world.

DPI-Nets combine three key features for effective particle-based simulation and control: multi-step spatial propagation, hierarchical particle structure, and dynamic interaction graphs. In particular, it employs dynamic interaction graphs, built on the fly throughout manipulation, to capture the meaningful interactions among particles of deformable objects and fluids. The use of dynamic graphs allows neural models to focus on learning meaningful interactions among particles, and is crucial for obtaining good simulation accuracy and high success rates in manipulation. As objects deform when robots interact with them, a fixed interaction graph over particles is insufficient for robot manipulating non-rigid objects.

Experiments demonstrate that DPI-Nets significantly outperform interaction networks (Battaglia et al., 2016), HRN (Mrowca et al., 2018), and a few other baselines. More importantly, unlike previous paper that focused purely on forward simulation, we have applied our model to downstream control tasks. Our DPI-Nets enable complex manipulation tasks for deformable objects and fluids, and adapts to scenarios with unknown physical parameters that need to be identified online. We have also performed real-world experiments to demonstrate our model's generalization ability.

## 2 RELATED WORK

**Differentiable physics simulators.** Researchers have developed many differentiable physical simulators (Ehrhardt et al., 2017; Degrave et al., 2016; Todorov et al., 2012), among which some can also provide analytical gradients (Tedrake & the Drake Development Team, 2019). In particular, Battaglia et al. (2016) and Chang et al. (2017) have explored learning a simulator from data by approximating object interactions with neural networks. Li et al. (2019) proposed learning to propagate signals along the interaction graph and extended to partially observable scenarios. These methods mostly focus on modeling rigid body dynamics.

Differentiable simulators for deformable objects have been less studied. Recently, Schenck & Fox (2018) proposed SPNets for differentiable simulation of position-based fluids (Macklin & Müller, 2013). An inspiring concurrent work from Mrowca et al. (2018) explored learning to approximate particle dynamics of deformable shapes with the Hierarchical Relation Network (HRN). Compared with these papers, we introduce state-specific modeling and dynamic graphs for accurate forward prediction for different states of matter (rigid bodies, deformable shapes, fluids). We also demonstrate how the learned dynamics model can be used for control in both simulation and real world.

Our approach is also complementary to some recent work on learning to discover the interaction graphs (van Steenkiste et al., 2018; Kipf et al., 2018). Our model can also be naturally augmented with a perception module to handle raw visual input, as suggested by Watters et al. (2017); Wu et al. (2017); Fragkiadaki et al. (2016).

**Model-predictive control with a differentiable simulator.** Many recent papers have studied model-predictive control with deep networks (Lenz et al., 2015; Gu et al., 2016; Nagabandi et al., 2018; Farquhar et al., 2018; Srinivas et al., 2018). They often learn an abstract state transition function, instead of an explicit account of the environment (Silver et al., 2017; Oh et al., 2017), and then use the learned function to facilitate training of a policy network. A few recent papers have employed analytical, differentiable simulators (de Avila Belbute-Peres et al., 2018; Schenck & Fox, 2018) for control problems, such as tool manipulation and tool-use planning (Toussaint et al., 2018). Our model builds on and extends these approaches by learning a general physics simulator that takes raw object observations (e.g., positions, velocities) of each particle as input. We then integrate it into classic trajectory optimization algorithms for control. Compared with pure analytical simulators, our learned simulator can better generalize to novel testing scenarios where object and environment parameters are unknown.

A few papers have explored using interaction networks for planning and control. They often learn a policy based on interaction networks' rollouts (Racanière et al., 2017; Hamrick et al., 2017; Pascanu et al., 2017). In contrast, our model learns a dynamics simulator and directly optimizes trajectories for continuous control. Recently, Sanchez-Gonzalez et al. (2018) have applied interaction networks for control, and Li et al. (2019) have further extended interaction nets to handle instance signal propagation for controlling multiple rigid bodies under partial observations. Compared with them, our dynamic particle interaction network simulate and control deformable, particle-based objects, using dynamic graphs to tackle scenes with complex object interactions.

## 3 APPROACH

### 3.1 PRELIMINARIES

We first describe how interaction networks (Battaglia et al., 2016) represent the physical system; we then extend them for particle-based dynamics.

The interactions within a physical system are represented as a directed graph, $G = \langle O, R \rangle$, where vertices $O = \{o_i\}$ represent objects and edges $R = \{r_k\}$ represent relations. Specifically, $o_i = \langle x_i, a_i^o \rangle$, where $x_i = \langle q_i, \dot{q}_i \rangle$ is the state of object $i$, containing its position $q_i$ and velocity $\dot{q}_i$. $a_i^o$ denotes its attributes (e.g., mass, radius). For relation, we have $r_k = \langle u_k, v_k, a_k^r \rangle, \quad 1 \leq u_k, v_k \leq |O|$, where $u_k$ is the receiver, $v_k$ is the sender, and both are integers. $a_k^r$ is the type and attributes of relation $k$ (e.g., collision, spring connection).

The goal is to build a learnable physical engine to capture the underlying physical interactions using function approximators $\phi$. The learned model can then be used to infer the system dynamics and predict the future from the current interaction graph as $G_{t+1} = \phi(G_t)$, where $G_t$ denotes the scene state at time $t$.

**Interaction networks.** Battaglia et al. (2016) proposed interaction networks (IN), a general-purpose, learnable physics engine that performs object- and relation-centric reasoning about physics. INs define an object function $f_O$ and a relation function $f_R$ to model objects and their relations in a compositional way. The future state at time $t + 1$ is predicted as $e_{k,t} = f_R(o_{u_k,t}, o_{v_k,t}, a_k^r)_{k=1...|R|}, \hat{o}_{i,t+1} = f_O(o_{i,t}, \sum_{k \in \mathcal{N}_i} e_{k,t})_{i=1...|O|}$, where $o_{i,t} = \langle x_{i,t}, a_i^o \rangle$ denotes

object $i$ at time $t$, $u_k$ and $v_k$ are the receiver and sender of relation $r_k$ respectively, and $\mathcal{N}_i$ denotes the relations where object $i$ is the receiver.

**Propagation networks.** A limitation of INs is that at every time step $t$, it only considers local information in the graph $G$ and cannot handle instantaneous propagation of forces, which however is a common phenomenon in rigid-body dynamics.

Li et al. (2019) proposed propagation networks to handle the instantaneous propagation of forces by doing multi-step message passing. Specifically, they first employed the ideas on fast training of RNNs (Lei & Zhang, 2017; Bradbury et al., 2017) to encode the shared information beforehand and reuse them along the propagation steps. The encoders for objects are denoted as $f_O^{\text{enc}}$ and the encoder for relations as $f_R^{\text{enc}}$, where we denote $c_{i,t}^o = f_O^{\text{enc}}(o_{i,t})$, $c_{k,t}^r = f_R^{\text{enc}}(o_{u_k,t}, o_{v_k,t}, a_k^r)$.

At time $t$, denote the propagating influence from relation $k$ at propagation step $l$ as $e_{k,t}^l$, and the propagating influence from object $i$ as $h_{i,t}^l$. For step $1 \leq l \leq L$, propagation can be described as

$$\text{Step 0:} \qquad h_{i,t}^0 = \mathbf{0}, \quad i = 1 \dots |O|, \tag{1}$$

$$\text{Step } l = 1, \dots, L: \qquad e_{k,t}^l = f_R(c_{k,t}^r, h_{u_k,t}^{l-1}, h_{v_k,t}^{l-1}), k = 1 \dots |R|, \tag{2}$$

$$h_{i,t}^l = f_O(c_{i,t}^o, \sum_{k \in \mathcal{N}_i} e_{k,t}^l, h_{i,t}^{l-1}), i = 1 \dots |O|, \tag{3}$$

$$\text{Output:} \qquad \hat{o}_{i,t+1} = f_O^{\text{output}}(h_{i,t}^L), \quad i = 1 \dots |O|, \tag{4}$$

where $f_O$ denotes the object propagatorand $f_R$ denotes the relation propagator.

## 3.2 DYNAMIC PARTICLE INTERACTION NETWORKS

Particle-based system is widely used in physical simulation due to its flexibility in modeling various types of objects (Macklin et al., 2014). We extend existing systems that model object-level interactions to allow particle-level deformation. Consider object set $\{\mathbf{o}_i\}$, where each object $\mathbf{o}_i = \{o_i^k\}_{k=1\dots|\mathbf{o}_i|}$ is represented as a set of particles. We now define the graph on the particles and the rules for influence propagation.

**Dynamic graph building.** The vertices of the graph are the union of particles for all objects $O = \{o_i^k\}_{i=1\dots|O|, k=1\dots|\mathbf{o}_i|}$. The edges $R$ between these vertices are dynamically generated over time to ensure efficiency and effectiveness. The construction of the relations is specific to environment and task, which we'll elaborate in Section 4. A common choice is to consider the neighbors within a predefined distance.

An alternative is to build a static, complete interaction graph, but it has two major drawbacks. First, it is not efficient. In many common physical systems, each particle is only interacting with a limited set of other particles (e.g., those within its neighborhood). Second, a static interaction graph implies a universal, continuous neural function approximator; however, many physical interactions involve discontinuous functions (e.g. contact). In contrast, using dynamic graphs empowers the model to tackle such discontinuity.

**Hierarchical modeling for long-range dependence.** Propagation networks Li et al. (2019) require a large $L$ to handle long-range dependence, which is both inefficient and hard to train. Hence, we add one level of hierarchy to efficiently propagate the long-range influence among particles (Mrowca et al., 2018). For each object that requires modeling of the long-range dependence (e.g. rigid-body), we cluster the particles into several non-overlapping clusters. For each cluster, we add a new particle as the cluster's root. Specifically, for each object $\mathbf{o}_i$ that requires hierarchical modeling, the corresponding roots are denoted as $\tilde{\mathbf{o}}_i = \{\tilde{o}_i^k\}_{k=1\dots|\tilde{\mathbf{o}}_i|}$, and the particle set containing all the roots is denoted as $\tilde{O} = \{\tilde{o}_i^k\}_{i=1\dots|O|, k=1\dots|\tilde{\mathbf{o}}_i|}$. We then construct an edge set $R_{\text{LeafToRoot}}$ that contains directed edges from each particle to its root, and an edge set $R_{\text{RootToLeaf}}$ containing directed edges from each root to its leaf particles. For each object that need hierarchical modeling, we add pairwise directed edges between all its roots, and denote this edge set as $R_{\text{RootToRoot}}$.

We employ a multi-stage propagation paradigm: first, propagation among leaf nodes, $\phi_{\text{LeafToLeaf}}(\langle O, R \rangle)$; second, propagation from leaf nodes to root nodes, $\phi_{\text{LeafToRoot}}(\langle O \cup$

$\tilde{O}, R_{\text{LeafToRoot}}\rangle)$; third, propagation between roots, $\phi_{\text{RootToRoot}}(\langle\tilde{O}, R_{\text{RootToRoot}}\rangle)$; fourth, propagation from root to leaf, $\phi_{\text{RootToLeaf}}(\langle O \cup \tilde{O}, R_{\text{RootToLeaf}}\rangle)$. The signals on the leaves are used to do the final prediction.

**Applying to objects of various materials.** We define the interaction graph and the propagation rules on particles for different types of objects as follows:

- *Rigid bodies.* All the particles in a rigid body are globally coupled; hence for each rigid object, we define a hierarchical model to propagate the effects. After the multi-stage propagation, we average the signals on the particles to predict a rigid transformation (rotation and translation) for the object. The motion of each particle is calculated accordingly. For each particle, we also include its offset to the center-of-mass to help determine the torque.

- *Elastic/Plastic objects.* For elastically deforming particles, only using the current position and velocity as the state is not sufficient, as it is not clear where the particle will be restored after the deformation. Hence, we include the particle state with the resting position to indicate the place where the particle should be restored. When coupled with plastic deformation, the resting position might change during an interaction. Thus, we also infer the motion of the resting position as a part of the state prediction. We use hierarchical modeling for this category but predict next state for each particles individually.

- *Fluids.* For fluid simulation, one has to enforce density and incompressibility, which can be effectively achieved by only considering a small neighborhood for each particle (Macklin & Müller, 2013). Therefore, we do not need hierarchical modeling for fluids. We build edges dynamically, connecting a fluid particle to its neighboring particles. The strong inductive bias leveraged in the fluid particles allows good performance even when tested on data outside training distributions.

For the interaction between different materials, two directed edges are generated for any pairs of particles that are closer than a certain distance.

### 3.3 Control on the Learned Dynamics

Model-based methods offer many advantages when comparing with their model-free counterparts, such as generalization and sample efficiency. However, for cases where an accurate model is hard to specify or computationally prohibitive, a data-driven approach that learns to approximate the underlying dynamics becomes useful.

Function approximators such as neural networks are naturally differentiable. We can rollout using the learned dynamics and optimize the control inputs by minimizing a loss between the simulated results and a target configuration. In cases where certain physical parameters are unknown, we can perform online system identification by minimizing the difference between the model's prediction and the reality. An outline of our algorithm can be found in Section A.

**Model predictive control using shooting methods.** Let's denote $\mathcal{G}_g$ as the goal and $\hat{u}_{1:T}$ be the control inputs, where $T$ is the time horizon. The control inputs are part of the interaction graph, such as the velocities or the initial positions of a particular set of particles. We denote the resulting trajectory after applying $\hat{u}$ as $\mathcal{G} = \{G_i\}_{i=1:T}$. The task here is to determine the control inputs as to minimize the distance between the actual outcome and the specified goal $\mathcal{L}_{\text{goal}}(\mathcal{G}, \mathcal{G}_g)$.

Our dynamic particle interaction network does forward simulation by taking the dynamics graph at time $t$ as input, and produces the graph at next time step, $\hat{G}_{t+1} = \Phi(G_t)$, where $\Phi$ is implemented as DPI-Nets as described in the previous section. Let's denote the the history until time $t$ as $\bar{\mathcal{G}} = \{G_i\}_{i=1...t}$, and the forward simulation from time step $t$ as $\hat{\mathcal{G}} = \{\hat{G}_i\}_{i=t+1...T}$. The loss $\mathcal{L}_{\text{goal}}(\bar{\mathcal{G}} \cup \hat{\mathcal{G}}, \mathcal{G}_g)$ can be used to update the control inputs by doing stochastic gradient descent (SGD). This is known as the shooting method in trajectory optimization (Tedrake, 2009).

The learned model might deviate from the reality due to accumulated prediction errors. We use Model-Predictive Control (MPC) (Camacho & Alba, 2013) to stabilize the trajectory by doing forward simulation and updating the control inputs at every time step to compensate the simulation error.

**Online adaptation.** In many real-world cases, without actually interacting with the environment, inherent attributes such as mass, stiffness, and viscosity are not directly observable. DPI-Nets can estimate these attributes on the fly with SGD updates by minimizing the distance between the predicted future states and the actual future states $\mathcal{L}_{\text{state}}(\hat{G}_t, G_t)$.

## 4 EXPERIMENTS

We evaluate our method on four different environments containing different types of objects and interactions. We will first describe the environments and show simulation results. We then present how the learned dynamics helps to complete control tasks in both simulation and the real world.

### 4.1 ENVIRONMENTS

**FluidFall** (Figure 2a). Two drops of fluids are falling down, colliding, and merging. We vary the initial position and viscosity for training and evaluation.

**BoxBath** (Figure 2b). A block of fluids are flushing a rigid cube. In this environment, we have to model two different materials and the interactions between them. We randomize the initial position of the fluids and the cube to test the model's generalization ability.

**FluidShake** (Figure 2c). We have a box of fluids and the box is moving horizontally, The speed of the box is randomly selected at each time step. We vary the size of the box and volume of the fluids to test generalization.

**RiceGrip** (Figure 2d). We manipulate an object with both elastic and plastic deformation (e.g., sticky rice). We use two cuboids to mimic the fingers of a parallel gripper, where the gripper is initialized at a random position and orientation. During the simulation of one grip, the fingers will move closer to each other and then restore to its original positions. The model has to learn the interactions between the gripper and the "sticky rice", as well as the interactions within the "rice" itself.

We use all four environments in evaluating our model's performance in simulation. We use the rollout MSE as our metric. We further use the latter two for control, because they involve fully actuated external shapes that can be used for object manipulation. In FluidShake, the control task requires determining the speed of the box at each time step, in order to make the fluid match a target configuration within a time window; in RiceGrip, the control task corresponds to select a sequence of grip configurations (position, orientation, closing distance) to manipulate the deformable object as to match a target shape. The metric for performance in control is the Chamfer distance between the manipulation results and the target configuration.

### 4.2 PHYSICAL SIMULATION

We present implementation details for dynamics learning in the four environment and perform ablation studies to evaluate the effectiveness of the introduced techniques.

**Implementation details.** For FluidFall, we dynamically build the interaction graph by connecting each particle to its neighbors within a certain distance $d$. No hierarchical modeling is used.

For BoxBath, we model the rigid cube as in Section 3.2, using multi-stage hierarchical propagation. Two directed edges will be constructed between two fluid particles if the distance between them is smaller than $d$. Similarly, we also add two directed edge between rigid particles and fluid particles when their distance is smaller than $d$.

For FluidShake, we model fluid as in Section 3.2. We also add five external particles to represent the walls of the box. We add a directed edge from the wall particle to the fluid particle when they are closer than $d$. The model is a single propagation network, where the edges are dynamically constructed over time.

For RiceGrip, we build a hierarchical model for rice and use four propagation networks for multi-stage effect propagation (Section 3.2). The edges between the "rice" particles are dynamically generated if two particles are closer than $d$. Similar to FluidShake, we add two external particles to represent the two "fingers" and add an edge from the "finger" to the "rice" particle if they are closer than the

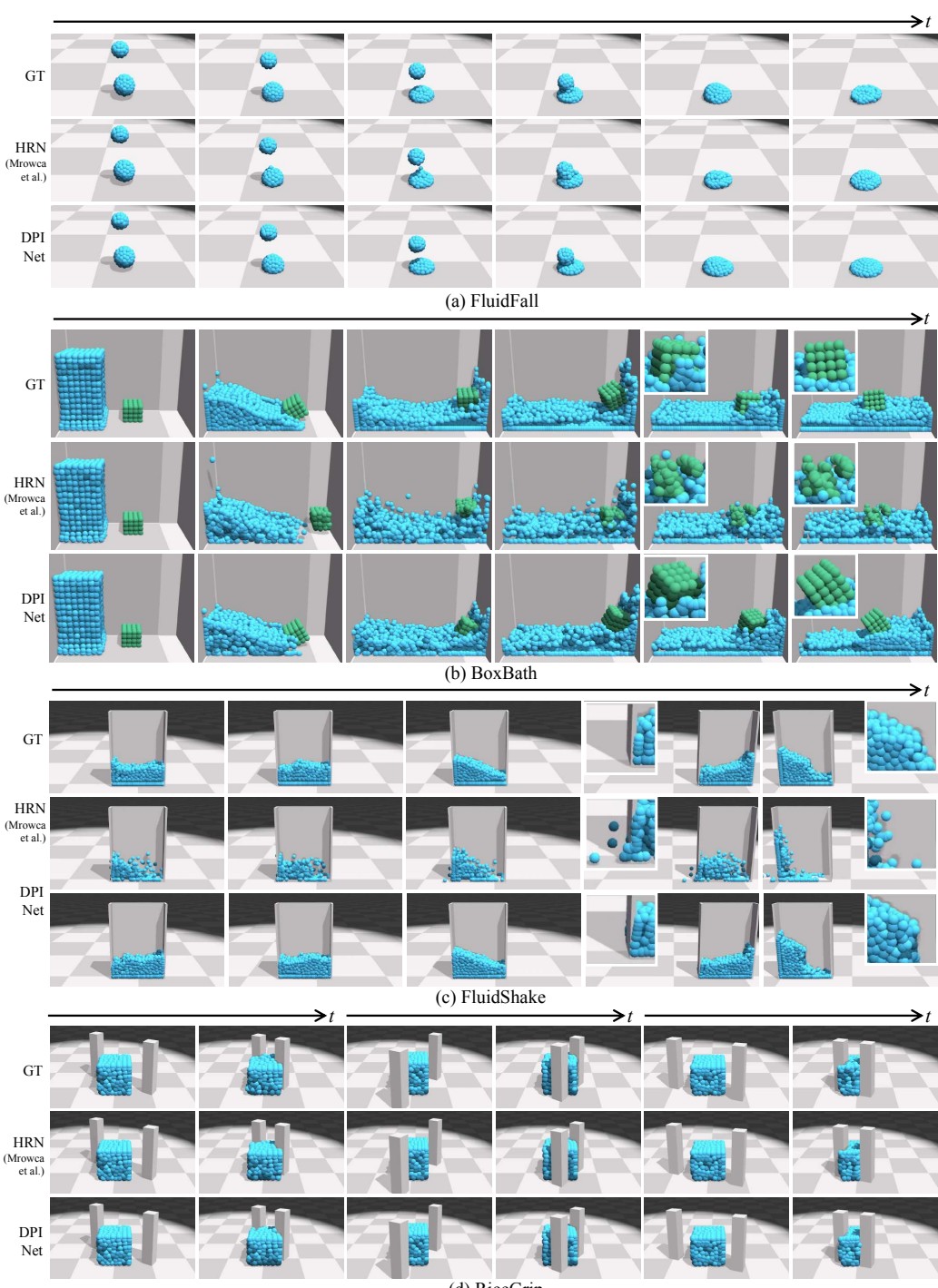

Figure 2: **Qualitative results on forward simulation.** We compare the ground truth (GT) and the rollouts from HRN (Mrowca et al., 2018) and our model (DPI-Net) in four environments (FluidFall, BoxBath, FluidShake, and RiceGrip). The simulations from our DPI-Net are significantly better. We provide zoom-in views for a few frames to show details. Please see our video for more empirical results.

distance $d$. As "rice" can deform both elastically and plastically, we maintain a resting position that helps the model restore a deformed particle. The output for each particle is a 6-dim vector for the velocity of the current observed position and the resting position. More training details for each environment can be found in Section D. Details for data generation are in Section C.

| Methods | FuildFall | BoxBath | FluidShake | RiceGrip |
|---|---|---|---|---|
| IN (Battaglia et al., 2016) | $2.74 \pm 0.56$ | N/A | N/A | N/A |
| HRN (Mrowca et al., 2018) | $0.21 \pm 0.04$ | $3.62 \pm 0.40$ | $3.58 \pm 0.77$ | $0.17 \pm 0.11$ |
| DPI-Net w/o hierarchy | $\mathbf{0.15 \pm 0.03}$ | $2.64 \pm 0.69$ | $\mathbf{1.89 \pm 0.36}$ | $0.29 \pm 0.13$ |
| DPI-Net | $\mathbf{0.15 \pm 0.03}$ | $\mathbf{2.03 \pm 0.41}$ | $\mathbf{1.89 \pm 0.36}$ | $\mathbf{0.13 \pm 0.07}$ |

Table 1: **Quantitative results on forward simulation.** MSE ($\times 10^{-2}$) between the ground truth and model rollouts. The hyperparameters used in our model are fixed for all four environments. FluidFall and FluidShake involve no hierarchy, so DPI-Net performs the same as the variant without hierarchy. DPI-Net significantly outperforms HRN (Mrowca et al., 2018) in modeling fluids (BoxBath and FluidShake) due to the use of dynamic graphs.

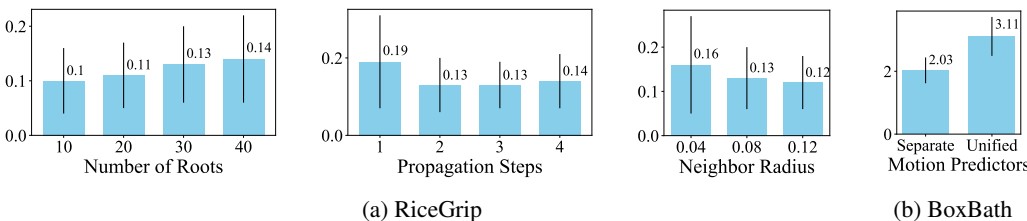

(a) RiceGrip                                                                    (b) BoxBath

Figure 3: **Ablation studies.** We perform ablation studies to test our model's robustness to hyperparameters. The performance is evaluated by the mean squared distance ($\times 10^{-2}$) between the ground truth and model rollouts. (a) We vary the number of roots when building hierarchy, the propagation step $L$ during message passing, and the size of the neighborhood $d$. (b) In BoxBath, DPI-Nets use separate motion predictors for fluids and rigid bodies. Here we compared with a unified motion predictor.

**Results.**  Qualitative and quantitative results are in Figure 2 and Table 1. We compare our method (DPI-Net) with three baselines, Interaction Networks (Battaglia et al., 2016), HRN (Mrowca et al., 2018), and DPI-Net without hierarchy. Note that we use the same set of hyperparameters in our model for all four testing environments.

Specifically, Interaction Networks (IN) consider a complete graph of the particle system. Thus, it can only operate on small environments such as FluidFall; it runs out of memory (12GB) for the other three environments. IN does not perform well, because its use of a complete graph makes training difficult and inefficient, and because it ignores influence propagation and long-range dependence.

Without a dynamic graph, modeling fluids becomes hard, because the neighbors of a fluid particle changes constantly. Table 1 shows that for environments that involve fluids (BoxBath and FluidShake), DPI-Net performs better than those with a static interaction graph. Our model also performs better in scenarios that involve objects of multiple states (BoxBath, Figure 2b), because it uses state-specific modeling. Models such as HRN (Mrowca et al., 2018) aim to learn a universal dynamics model for all states of matter. It is therefore solving a harder problem and, for this particular scenario, expected to perform not as well. When augmented with state-specific modeling, HRN's performance is likely to improve, too. Without hierarchy, it is hard to capture long-range dependence, leading to performance drop in environments that involve hierarchical object modeling (BoxBath and RiceGrip).

Appendix B includes results on scenarios outside the training distribution (e.g., more particles). DPI-Net performs well on these out-of-sample cases, successfully leveraging the inductive bias.

**Ablation studies.**  We also test our model's sensitivity to hyperparameters. We consider three of them: the number of roots for building hierarchy, the number of propagation steps $L$, and the size of the neighborhood $d$. We test them in RiceGrip. As can be seen from the results shown in Figure 3a, DPI-Nets can better capture the motion of the "rice" by using fewer roots, on which the information might be easier to propagate. Longer propagation steps do not necessarily lead to better performance, as they increases training difficulty. Using larger neighborhood achieves better results, but makes computation slower. Using one TITAN Xp, each forward step in RiceGrip takes 30ms for $d = 0.04$, 33ms for $d = 0.08$, and 40ms for $d = 0.12$.

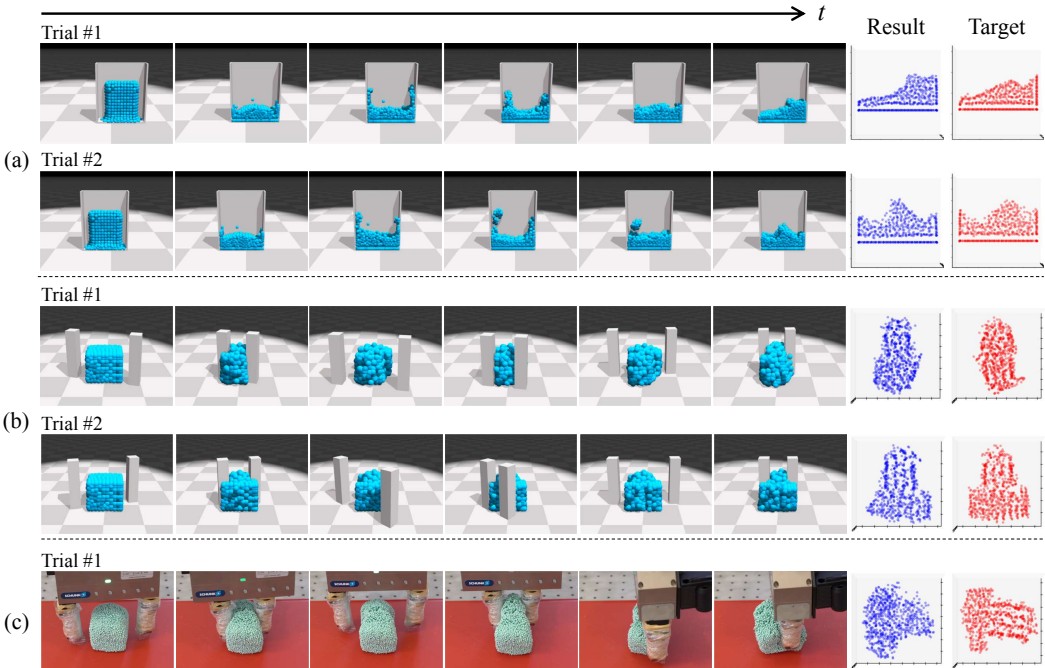

Figure 4: **Qualitative results on control.** (a) FluidShake - Manipulating a box of fluids to match a target shape. The Result and Target indicate the fluid shape when viewed from the cutaway view. (b) RiceGrip - Gripping a deformable object and molding it to a target shape. (c) RiceGrip in Real World - Generalize the learned dynamics and the control algorithms to the real world by doing online adaptation. The last two columns indicate the final shape viewed from the top.

We also perform experiments to justify our use of different motion predictors for objects of different states. Figure 3b shows the results of our model *vs.* a unified dynamics predictor for all objects in BoxBath. As there are only a few states of interest (solids, liquids, and soft bodies), and their physical behaviors are drastically different, it is not surprising that DPI-Nets, with state-specific motion predictors, perform better, and are equally efficient as the unified model (time difference smaller than 3ms per forward step).

## 4.3 CONTROL

We leverage dynamic particle interaction network for control tasks in both simulation and real world. Because trajectory optimization using shooting method can easily stuck to a local minimum, we first randomly sample $N_{sample}$ control sequences, and select the best performing one according to the rollouts of our learned model. We then optimize it via shooting method using our model's gradients. We also use online system identification to further improve the model's performance. Figure 4 and Figure 5 show qualitative and quantitative results, respectively. More details of the control algorithm can be found in Section E.

**FluidShake.** We aim to control the speed of the box to match the fluid particles to a target configuration. We compare our method (RS+TO) with random search over the learned model (without trajectory optimization - RS) and Model-free Deep Reinforcement Learning (Actor-Critic method optimized with PPO (Schulman et al., 2017) (RL). Figure 5a suggests that our model-based control algorithm outperforms both baselines with a large margin. Also RL is not sample-efficient, requiring more than 10 million time steps to converge while ours requires 600K time steps.

**RiceGrip.** We aim to select a sequence of gripping configurations (position, orientation, and closing distance) to mold the "sticky rice" to a target shape. We also consider cases where the stiffness of the rice is unknown and need to be identified. Figure 5b shows that our dynamic particle interaction

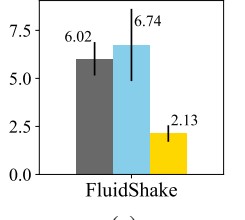 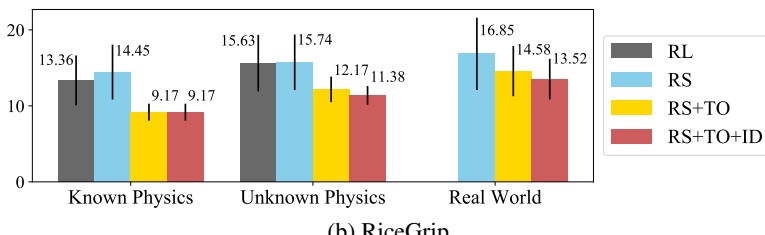

(a)                                       (b) RiceGrip

Figure 5: **Quantitative results on control.** We show the results on control (as evaluated by the Chamfer distance ($\times 10^{-2}$) between the manipulated result and the target) for (a) FluidShake and (b) RiceGrip by comparing with four baselines. **RL**: Model-free deep reinforcement learning optimized with PPO; **RS**: Random search the actions from the learned model and select the best one to execute; RS + **TO**: Trajectory optimization augmented with model predictive control; RS + TO + **ID**: Online system identification by estimating uncertain physical parameters during run time.

network with system identification performs the best, and is much more efficient than RL (150K *vs.* 10M time steps).

**RiceGrip in the real world.** We generalize the learned model and control algorithm for RiceGrip to the real world. We first reconstruct object geometry using a depth camera mounted on our Kuka robot using TSDF volumetric fusion (Curless & Levoy, 1996). We then randomly sampled $N_{\text{fill}}$ particles within the object mesh as the initial configuration for manipulation. Figure 4c and Figure 5b shows that, using DPI-Nets, the robot successfully adapts to the real world environment of unknown physical parameters and manipulates a deformable foam into various target shapes. The learned policy in RiceGrip does not generalize to the real world due to domain discrepancy, and outputs invalid gripping configurations.

## 5   CONCLUSION

We have demonstrated that a learned particle dynamics model can approximate the interaction of diverse objects, and can help to solve complex manipulation tasks of deformable objects. Our system requires standard open-source robotics and deep learning toolkits, and can be potentially deployed in household and manufacturing environment. Robot learning of dynamic scenes with particle-based representations shows profound potentials due to the generalizability and expressiveness of the representation. Our study helps lay the foundation for it.

## ACKNOWLEDGEMENT

This work was supported by: Draper Laboratory Incorporated, Sponsor Award No. SC001-0000001002; NASA - Johnson Space Center, Sponsor Award No. NNX16AC49A; NSF #1524817; DARPA XAI program FA8750-18-C000; ONR MURI N00014-16-1-2007; Facebook.

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

## A    CONTROL ALGORITHM

---

**Algorithm 1** Control on Learned Dynamics at Time Step $t$

---

    **Input:** Learned forward dynamics model $\Phi$
        Predicted dynamics graph $\hat{G}_t$
        Current dynamics graph $G_t$
        Goal $\mathcal{G}_g$, current estimation of the attributes $A$
        Current control inputs $\hat{u}_{t:T}$
        States history $\bar{\mathcal{G}} = \{G_i\}_{i=1...t}$
        Time horizon $T$
    **Output:**  Controls $\hat{u}_{t:T}$, predicted next time step $\hat{G}_{t+1}$

    Update $A$ by descending with the gradients $\nabla_A \mathcal{L}_{\text{state}}(\hat{G}_t, G_t)$
    Forward simulation using the current graph $\hat{G}_{t+1} \leftarrow \Phi(G_t)$
    Make a buffer for storing the simulation results $\mathcal{G} \leftarrow \bar{\mathcal{G}} \cup \hat{G}_{t+1}$
    **for** $i = t + 1, ..., T - 1$ **do**
        Forward simulation: $\hat{G}_{j+1} \leftarrow \Phi(\hat{G}_j); \mathcal{G} \leftarrow \mathcal{G} \cup \hat{G}_{j+1}$
    **end for**
    Update $\hat{u}_{t:T}$ by descending with the gradients $\nabla_{\hat{u}_{t:T}} \mathcal{L}_{\text{goal}}(\mathcal{G}, \mathcal{G}_g)$
    Return $\hat{u}_{t:T}$ and $\hat{G}_{t+1} \leftarrow \Phi(G_t)$

---

## B    GENERALIZATION ON EXTRAPOLATION

We show our model's performance on fluids, rigid bodies, and deformable objects with a larger number of particles than they have in the training set. Figure 6 shows qualitative and quantitative results. Our model scales up well to larger objects.

## C    DATA GENERATION

The data is generated using NVIDIA FleX. We have developed a Python interface for the ease of generating and interacting with different environments. We will release the code upon publication.

**FluidFall.**    We generated 3,000 rollouts over 120 time steps. The two drops of fluids contain 64 and 125 particles individually, where the initial position of one of the drop in the 3 dimensional coordinates is uniformly sampled between (0.15, 0.55, 0.05) and (0.25, 0.7, 0.15), while the other drop is uniformly sampled between (0.15, 0.1, 0.05) and (0.25, 0.25, 0.15).

**BoxBath.**    We generated 3,000 rollouts over 150 time steps. There are 960 fluid particles and the rigid cube consist particles ranging between 27 and 150. The fluid particle block is initialized at (0, 0, 0), and the initial position of the rigid cube is randomly initialized between (0.45, -0.0155, 0.02) to (1.2, -0.0155, 0.4).

**FluidShake.**    We generated 2,000 rollouts over 300 time steps. The height of the box is 1.0 and the thickness of the wall is 0.025. For the initial fluid cuboid, the number of fluid particles is uniformly sampled between 10 and 12 in the x direction, between 15 and 20 in the y direction, 3 in the z direction. The box is fixed in the y and z direction, and is moving freely in the x direction. We randomly place the initial x position between -0.2 to 0.2. The sampling of the speed is implemented as $v = v + \text{rand}(-0.15, 0.15) - 0.1x$, in order to encourage motion smoothness and moving back to origin, where speed v is initialized as 0.

**RiceGrip.**    We generated 5,000 rollouts over 30 time steps. We randomize the size of the initial "rice" cuboid, where the length of the three sides is uniformly sampled between 8.0 and 10.0. The material property parameters `clusterStiffness` is uniformly sampled between 0.3 and 0.7, `clusterPlasticThreshold` is uniformly sampled between 1e-5 and 5e-4, and

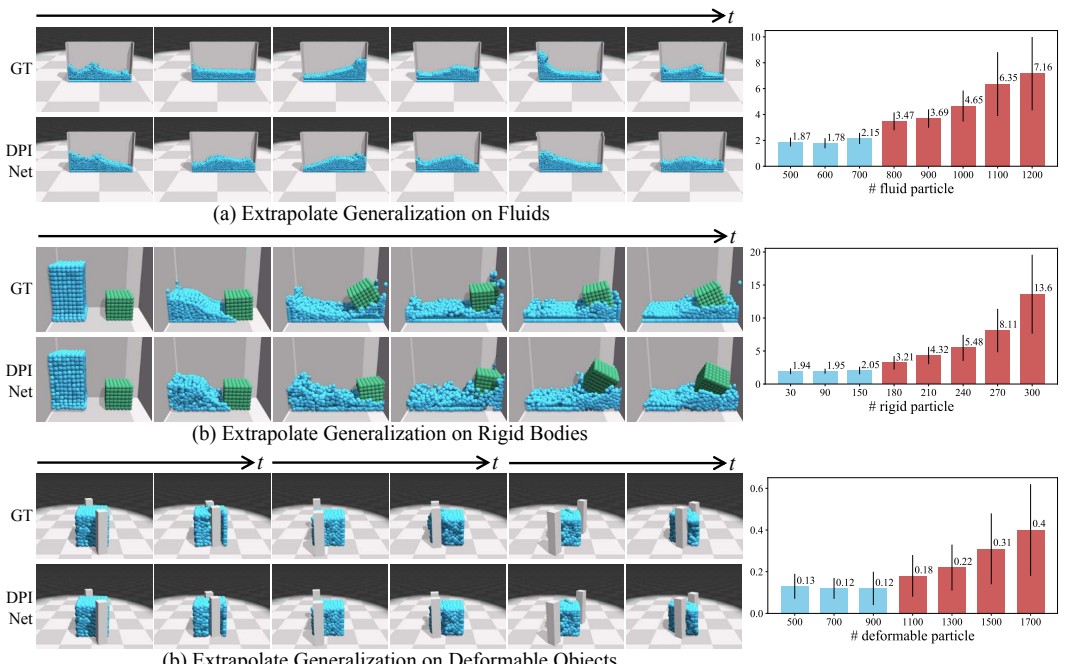

Figure 6: **Extrapolate generalization on fluids, rigid bodies, and deformable objects.** The performance is evaluated by the MSE ($\times 10^{-2}$) between the ground truth and rollouts from DPI-Nets. The blue bars denote the range of particle numbers that have been seen during training, which indicate interpolation performance. The red bars indicate extrapolation performance that our model can generalize to cases containing two times more particles than cases it has been trained on.

`clusterPlasticCreep` is uniformly sampled between 0.1 and 0.3. The position of the gripper is randomly sampled within a circle of radius 0.5. The orientation of the gripper is always perpendicular to the line connecting the origin to the center of the gripper and the close distance is uniformly sampled between 0.7 to 1.0.

Of all the generated data, 90% of the rollouts are used for training, and the rest 10% are used for validation.

## D  TRAINING DETAILS

The models are implemented in PyTorch, and are trained using Adam optimizer (Kingma & Ba (2015)) with a learning rate of 0.0001. The number of particles and relations might be different at each time step, hence we use a batch size of 1, and we update the weights of the networks once every 2 forward rounds.

The neighborhood $d$ is set as 0.08, and the propagation step $L$ is set as 2 for all four environments. For hierarchical modeling, it does not make sense to propagate more than one time between leaves and roots as they are disjoint particle sets, and each propagation stage between them only involves one-way edges; hence $\phi_{\text{LeafToLeaf}}$ uses $L = 2$. $\phi_{\text{LeafToRoot}}$ uses $L = 1$. $\phi_{\text{RootToRoot}}$ uses $L = 2$, and $\phi_{\text{RootToLeaf}}$ uses $L = 1$.

For all propagation networks used below, the object encoder $f_O^{\text{enc}}$ is an MLP with two hidden layers of size 200, and outputs a feature map of size 200. The relation encoder $f_R^{\text{enc}}$ is an MLP with three hidden layers of size 300, and outputs a feature map of size 200. The propagator $f_O$ and $f_R$ are both MLP with one hidden layer of size 200, in which a residual connection is used to better propagate the effects, and outputs a feature map of size 200. The propagators are shared within each stage of propagation. The motion predictor $f_O^{\text{output}}$ is an MLP with two hidden layers of size 200, and output the state of required dimension. ReLU is used as the activation function.

**FluidFall.** The model is trained for 13 epochs. The output of the model is the 3 dimensional velocity, which is multiplied by $\Delta t$ and added to the current position to do rollouts.

**BoxBath.** In this environment, four propagation networks are used due to the hierarchical modeling and the number of roots for the rigid cube is set as 8. We have two separate motion predictor for fluids and rigid body, where the fluid predictor output velocity for each fluid particle, while the rigid predictor takes the mean of the signals over all its rigid particles as input, and output a rigid transformation (rotation and translation). The model is trained for 5 epochs.

**FluidShake.** Only one propagation network is used in this environment, and the model is trained for 5 epochs.

**RiceGrip.** Four propagation networks are used due to the hierarchical modeling, and the number of roots for the "rice" is set as 30. The model is trained for 20 epochs.

## E  CONTROL DETAILS

$N_{\text{sample}}$ is chosen as 20 for all three cases, where we sample 20 random control sequences, and choose the best performing one as evaluated using our learned model. The evaluation is based on the Chamfer distance between the controlling result and the target configuration.

**FluidShake.** In this environment, the control sequence is the speed of the box along the x axis. The method to sample the candidate control sequence is the same as when generating training data of this environment. After selected the best performing control sequence, we first use RMSprop optimizer to optimize the control inputs for 10 iterations using a learning rate of 0.003. We then use model-predictive control to apply the control sequence to the FleX physics engine using Algorithm 1.

**RiceGrip.** In this environment, we need to come up with a sequence of grip configurations, where each grip contains positions, orientation, and closing distance. The method to sample the candidate control sequence is the same as when generating training data of this environment. After selected the best performing control sequence, we first use RMSprop optimizer to optimize the control inputs for 20 iterations using a learning rate of 0.003. We then use model-predictive control to apply the control sequence to the FleX physics engine using Algorithm 1.

**RiceGrip in Real World.** In this environment, we need to come up with a sequence of grip configurations, where each grip contains positions, orientation, and closing distance. The method to sample the candidate control sequence is the same as when generating training data of **RiceGrip**, and $N_{\text{fill}}$ is chosen as 768. Different from the previous case, the physical parameters are always unknown and has to be estimated online. After selected the best performing control sequence, we first use RMSprop optimizer to optimize the control inputs for 20 iterations using a learning rate of 0.003. We then use model-predictive control to apply the control sequence to the real world using Algorithm 1.

