# OpenReview forum: "Learning Particle Dynamics for Manipulating Rigid Bodies, Deformable Objects, and Fluids"
_ICLR.cc/2019/Conference_

### Official Review · AnonReviewer2 · 2018-11-02
**Impressive work**

**Rating:** 7
**Confidence:** 3

**Review:**

This work demonstrates that a particle dynamics model can be learned to approximate the interaction of various objects. The resulting differentiable simulator has a strong inductive bias, which makes it possible to efficiently solve complex manipulation tasks over deformable objects.

# Quality

This work is an impressive proof-of-concept of the capabilities of differentiable programming for learning complex (physical) processes, such as particle dynamics. In my opinion, the resulting particle interaction network would deserve publication for itself. However, this work goes already one step further and demonstrates that the resulting differentiable simulator can be used for the manipulation of deformable objects.

The method is evaluated on a well-rounded set of experiments which demonstrates its potential. More real-world experiments would be welcome to leave any doubt.

EDIT: However, the current manuscript lacks a proper comparison with (cited) previous work, such as 1806.08047.

# Clarity

The paper is well written, although I do feel it was difficult to remain within the 8-page limit given the breadth of the work.

# Originality

As far as I know, this work is (very) original. (That being said, I am not too familiar with the related work.)

EDIT: This work is actually quite similar to 1806.08047. A proper discussion of the differences should be included.

# Significance

This work will certainly be of interest for several research communities, including deep learning, physics, control and robotics.

---

> ### Author Response · Authors · 2018-11-20
> **Response to Reviewer 2**
>
> Thank you so much for the encouraging comments!
>
> Our paper is novel, because it first explores learning particle dynamics among fluids and rigid bodies, with a dynamically-built graph, and because it first applies the learned particle dynamics model for control, in both synthetic environments and on a real robot. There are also key differences between our paper and the inspiring early work from Mrowca et al. [1]:
> 1) We learn dynamics between particles that are in different physical states (e.g. between rigid bodies and fluids), while Mrowca et al. [1] focused on scenarios where all objects are in the same state, e.g. all soft or rigid bodies.
> 2) While Mrowca et al. used a static graph, our model uses an interaction graph built dynamically. Since maintaining a dynamic graph is crucial for simulating objects undergoing large deformation like fluids, this modification has enabled our model to work better in the more challenging cases mentioned above.
> 3) We have applied our model to more challenging control problems, including one on a real-world robot. In the other paper (Mrowca et al. [1]), however, this non-trivial task was not explored at all. Demonstrating the power of learned particle dynamics models on downstream control tasks is an important contribution to the learning and robotics community. It should not be undervalued.
>
> We agree that it’ll be important to clearly discuss the differences and include additional comparisons. In our revision by Nov. 26, we will compare with Mrowca et al. [1] in all four environments we used, contrasting the two models’ capacity in simulating rigid bodies, elastic deformation, and fluids.
>
> We have also listed all other planned changes in our general response above. Please don’t hesitate to let us know for any additional comments on the paper or on the planned changes.
>
> [1] Damian Mrowca, Chengxu Zhuang, Elias Wang, Nick Haber, Li Fei-Fei, Joshua B. Tenenbaum, Daniel L. K. Yamins. "Flexible Neural Representation for Physics Prediction." In NIPS, 2018.

---

### Official Review · AnonReviewer1 · 2018-11-04
**Similar to Interaction Networks; Missing Comparisons to Baselines**

**Rating:** 6
**Confidence:** 4

**Review:**

[Paper Summary]
This paper tackles the problem of learning dynamics of non-rigid objects in a physics simulator. This learned dynamics can then be used for planning later. The non-rigid objects are represented via a particle-based system. The dynamics model is learned using NVIDIA's particle-based simulator "Flex". The main idea is to adapt Interaction Networks [Battaglia, 2016] which was earlier proposed for rigid-body simulators to particle-based simulators. Instead of maintaining interactions at the level of objects as in [Battaglia, 2016], the proposed approach models interaction at the level of particles.

[Paper Strengths]
The paper is clearly written and tackles an important research problem. The existing literature is presented well.

[Paper Weaknesses]
=> The introduction and the text in the first two pages seem to be introducing a new way to model "dynamic" interactions between particles for handling non-rigid transformations. However, upon reading the method section, the approach seems to be a direct application of the Interaction Graph Networks (originally applied to the rigid-body simulator) to the particle-based simulator. The only difference is that instead of maintaining a fully-connected graph (memory and computational bottleneck), each particle is only connected to the near-by particles within distance d.

=> One of the major issue with the paper is the experimental section of the paper. Since the proposed method is quite incremental over the prior work, a strong empirical section is must to justify the approach. Here are the comments:
   - Since the proposed approach is an adaptation of [Battaglia, 2016], it should be compared to other existing methods. The experiment section in its current state does not compare to any baseline. The well-written related work (section-2) talks about (Mrowca et.al. 2018) and (Schenck and Fox, 2018) as the works which investigate learning dynamics of deformable objects using a particle-based simulator. However, no comparison is provided to either of the methods. Hence, it is not possible to judge the quality of the presented results.

   - All results in Figure-5 or Figure-3 are quite close to each other. It is not clear whether the improvement is significant or not since the error bars are not provided at all.

   - No ablation is performed to test the sensitivity of the proposed method with respect to the hyper-parameters introduced; for instance, the distance 'd'.

=> The name "Dynamic Particle Interaction" is overloaded with terms, especially, the use of word 'dynamic' here just refers to the interaction of particles to model deformable objects. This "dynamic" interaction is not "learned" but simply hard-coded by deleting the edges which are farther than d distance apart and adding near ones. Something like "Particle-level Interaction Networks" would be a more honest description of the approach.

[Final Recommendation]
I request the authors to address the comments raised above and will decide my final rating based on that. With the current set of experiments, the paper doesn't seem to be ready yet.

---

> ### Author Response · Authors · 2018-11-20
> **Response to Reviewer 1**
>
> Thank you so much for your insightful and detailed comments!
>
> 1. Our contributions and comparisons with baselines
> Thanks for the suggestions. Our paper is novel, because it first explores learning particle dynamics among fluids and rigid bodies, with a dynamically-built graph, and because it first applies the learned particle dynamics model for control, in both synthetic environments and on a real robot. Here, we summarize the three key differences between our paper and earlier works that serve as our inspiration [1, 2]:
>
> 1) We learn dynamics between particles that are in different physical states (e.g. between rigid bodies and fluids), while other works focused on scenarios where all objects are in the same state, e.g. all soft or rigid bodies (Mrowca et al. [1]), or all fluids (Schenck and Fox [2]).
> 2) While Mrowca et al. used a static graph, our model uses an interaction graph built dynamically. Since maintaining a dynamic graph is crucial for simulating objects undergoing large deformation like fluids, this modification has enabled our model to work better in the more challenging cases mentioned above.
> 3) We have applied our model to more challenging control problems, including one on a real-world robot. In the other paper (Mrowca et al. [1]), however, this non-trivial task was not explored at all. Demonstrating the power of learned particle dynamics models on downstream control tasks is an important contribution to the learning and robotics community. It should not be undervalued.
>
> Our dynamics model includes many improvements compared with the vanilla interaction networks: the use of dynamic graphs, multi-step propagations, and the hierarchical structure. While the hierarchical structure and multi-step propagations have been explored (Mrowca et al and Li et al), dynamically-built interaction graphs are new, the combination of these ideas is new, and most importantly, applying them to control is new.
>
> In particular, a dynamically-built graph can better capture the inductive bias for many common physical systems like fluids and gases, as well as extrapolations to unseen environments (Fig. 3). We agree with the reviewer that learning to build dynamic graphs is definitely an important future direction, which we’re currently pursuing, but we’d also like to emphasize that the idea of updating the interaction graph based on the states of the particles is conceptually important and empirically useful.
>
> We agree that it’ll be important to clearly discuss the differences and include additional comparisons. In our revision by Nov. 26, we will compare with Mrowca et al. [1] in all four environments we used, contrasting the two models’ capacity in simulating rigid bodies, elastic deformation, and fluids.
>
> 2. Ablation studies
> We’ll add confidence intervals and error bars to Fig. 3 and 5. The improvements of the simulation results in Fig. 3 are significant: on average, the rollout loss decreases by 85% with the use of dynamic graphs. We will also add more comparisons with related works demonstrating the effectiveness of our introduced techniques.
>
> The hyperparameters used in the paper are not specific to environments. The supplementary material shows that most hyperparameters are kept the same across scenarios. We will provide systematic analyses regarding the sensitivity of the hyperparameters in the revision, for
> 1) the propagation step L,
> 2) the number of roots in the rigid/deformable bodies, and
> 3) the neighborhood distance d.
>
> We have also listed all other planned changes in our general response above. Please don’t hesitate to let us know for any additional comments on the paper or on the planned changes.
>
> [1] Damian Mrowca, Chengxu Zhuang, Elias Wang, Nick Haber, Li Fei-Fei, Joshua B. Tenenbaum, Daniel L. K. Yamins. "Flexible Neural Representation for Physics Prediction." In NIPS, 2018.
> [2] Connor Schenck, Dieter Fox. “SPNets: Differentiable Fluid Dynamics for Deep Neural Networks.” In CoRL 2018.

---

> > ### Comment · AnonReviewer1 · 2018-12-08
> > **Follow-up on the rebuttal**
> >
> > I went through the author's reply and the updated paper in detail. The major part of the authors' response above mainly argues about the differences of proposed method from Mrowca et.al. 2018. However, this is neither the question I asked nor had any concerns about.
> >
> > My main concern is that the approach seems to be a direct application of the Interaction Graph Networks to the particle-based simulator. The only difference is that instead of maintaining a fully-connected graph, each particle is only connected to the near-by particles within distance d. However, the paper is written in a way which lefts the reader wonder about what is the new concept that is being introduced in "Dynamic Particle Interaction" networks. The authors' response above or the updated paper did not acknowledge this as an issue.
> >
> > The authors have added error bars in Fig-3,5 which basically show that the results are within the error bars and hence not significant.
> >
> > However, I appreciate the to Mrowca et.al. 2018 and updating my rating from 5 to 6. But I still believe that the paper should tone down the emphasis on the "dynamic" graph part.

---

> ### Author Response · Authors · 2018-12-03
> **Looking forward to your feedback**
>
> Dear Reviewer 1,
>
> Thanks again for your constructive comments. We have made substantial changes in the revision according to your review. In particular, we’ve included detailed comparisons with (Mrowca et al, 2018), ablation studies, and errors bars. As the discussion period is about to end, please don’t hesitate to let us know if there are any additional clarifications that we can offer, as we would love to convince you of the merits of the paper. Thanks!

---

### Official Review · AnonReviewer3 · 2018-11-04
**Little Novelty over Prior Work**

**Rating:** 8
**Confidence:** 3

**Review:**

The authors present an algorithm for learning the dynamics prediction of deformable and fluid bodies by modeling them as (potentially hierarchical) systems of many interacting particles.  This model applies a shared encoder to the particle states (positions and velocities), a shared relation network to nearby pairs, and a shared propagator network to the summed relation network outputs.  In some cases this process is applied in a multi-scale hierarchical fashion.  The authors demonstrate accurate rollouts of system dynamics and usefulness for manipulative control of deformable objects.

I find the motivation in the introduction persuasive and the algorithmic approach sound.  I also like the application to RL.  However, I do have some concerns, as follows:

1)  The novelty of the method is questionable.  Specifically, the hierarchical interaction network proposed here seems extremely similar to the prior (and cited) paper (Mrowca et al., 2018), which the authors do not directly compare against.  If there is a non-negligible difference between the two algorithms, then the authors should explicitly discuss the difference and empirically compare the two, in order to benefit others in the community who otherwise would not know which to use.

2)  The paper would benefit a lot from a diagram of the model.  Specifically, it would be good to have a diagram of the hierarchical interaction network demonstrating the multiscale propagation.  This could go in Figure 1, perhaps replacing elements (b) and (d) of the current Figure 1, which in my opinion are unnecessary and can be removed.

3)  The paper uses domain-specific hyperparameters, yet does not discuss or analyze the effects of these hyperparameters much.  Specifically, for this method to be useful to others, we would like to know how to choose (i) the propagation step L, (ii) the number of roots, and (iii) the neighborhood distance d.  In the paper, these numbers are chosen differently for the different environments without explanation.  Graphs showing performance on each task over a range of values of these parameters would be good (perhaps in the supplementary material).  Also, using the same hyperparameters for all environments (or at least a common generating function) would help support the generality of this model.

4)  The treatment of rigid bodies seems a bit hand-held.  Specifically, to determine the dynamics of rigid bodies, there is a ground-truth calculation which calculates computes the velocity and angular velocity of the body from the model predictions for its constituent particles.  Furthermore, if I understand correctly, there is a different motion predictor network for those particles in a rigid body than those in the surrounding fluid --- is this correct?  If so, this raises the questions:  (i) What happens if the same motion-predictor network is used for all particles, and (ii) What happens if the ground-truth rigid dynamics calculation is not done, so the model has to do all the work?  It would be interesting to have these as baselines.

5)  It would be nice to see more generalization results.  There is only one generalization experiment, testing for generalization over particle number in FluidShake.  However, the FluidShake model is not hierarchical.  The hierarchical models are a big emphasis in the paper, so showing generalization on BoxBath or RiceGrip would be much more meaningful.

6)  No confidence intervals for the quantitative results.  Confidence intervals would be good to see in the table in Figure 3-a.  Also, the bar graph in Figure 5 really would benefit from errorbars --- it is difficult to determine if the results are significant.

7)  While the text is generally clear and definitely understandable, I have a couple of comments about it:  (i) The last three paragraphs of the introduction are repetitive and I think they can be removed, or at least shortened a lot.  There are also quite a number of grammatical errors throughout the paper, though it is still comprehensible.

EDIT:  In their revision the authors addressed these concerns well and the paper is much more convincing (see longer comment below).  In light of this I have changed my rating from a 5 to an 8.

---

> ### Author Response · Authors · 2018-11-20
> **Response to Reviewer 3**
>
> Thank you so much for the insightful and detailed comments!
>
> 1. Novelty and Baselines
> Thanks for the suggestions. Our paper is novel, because it first explores learning particle dynamics among fluids and rigid bodies, with a dynamically-built graph, and because it first applies the learned particle dynamics model for control, in both synthetic environments and on a real robot. Here, we summarize the three key differences between our paper and earlier works that serve as our inspiration [1, 2]:
>
> 1) We learn dynamics between particles that are in different physical states (e.g. between rigid bodies and fluids), while other works focused on scenarios where all objects are in the same state, e.g. all soft or rigid bodies (Mrowca et al. [1]), or all fluids (Schenck and Fox [2]).
> 2) While Mrowca et al. used a static graph, our model uses an interaction graph built dynamically. Since maintaining a dynamic graph is crucial for simulating objects undergoing large deformation like fluids, this modification has enabled our model to work better in the more challenging cases mentioned above.
> 3) We have applied our model to more challenging control problems, including one on a real-world robot. In the other paper (Mrowca et al. [1]), however, this non-trivial task was not explored at all. Demonstrating the power of learned particle dynamics models on downstream control tasks is an important contribution to the learning and robotics community. It should not be undervalued.
>
> We agree that it’ll be important to clearly discuss the differences and include additional comparisons. In our revision by Nov. 26, we will compare with Mrowca et al. [1] in all four environments we used, contrasting the two models’ capacity in simulating rigid bodies, elastic deformation, and fluids.
>
> 2. Presentation
> Thanks. As suggested, we will update figure 1 to include a diagram of the model demonstrating our key contribution: the joint use of dynamic graphs, hierarchical structure, and multi-step propagation. We’ll add confidence interval and error bars and revise the introduction.
>
> 3. Rigid body representation
> State-specific dynamics models perform better and are equally efficient. We use state-specific models because there are only a few states of interest (solids, liquids, and soft bodies), and their physical behaviors are drastically different. We agree with the reviewer that additional comparisons are important: we’ll add experiments with a unified dynamics model.
>
> The motion of a rigid body only has six degrees of freedom. Predicting per-particle movement does not work well, compared with predicting a global rigid motion, because particles will deform and scatter. We will also add a comparison to the revision.
>
> 4. Generalization
> We will show more (extrapolate) generalization results in RiceGrip and BoxBath to demonstrate that our method can generalize to novel environments that are larger than the training ones.
>
> 5. Hyperparameters
> The hyperparameters used in the paper are not specific to environments. The supplementary material shows that most hyperparameters are kept the same across scenarios. We will provide systematic analyses regarding the sensitivity of the hyperparameters in the revision, for
> 1) the propagation step L,
> 2) the number of roots in the rigid/deformable bodies, and
> 3) the neighborhood distance d.
>
> We have also listed all other planned changes in our general response above. Please don’t hesitate to let us know for any additional comments on the paper or on the planned changes.
>
> [1] Damian Mrowca, Chengxu Zhuang, Elias Wang, Nick Haber, Li Fei-Fei, Joshua B. Tenenbaum, Daniel L. K. Yamins. "Flexible Neural Representation for Physics Prediction." In NIPS, 2018.
> [2] Connor Schenck, Dieter Fox. “SPNets: Differentiable Fluid Dynamics for Deep Neural Networks.” In CoRL 2018.

---

> > ### Comment · AnonReviewer3 · 2018-11-27
> > **The Revision is a Significant Improvement**
> >
> > Dear Authors,
> >
> > Thank you very much for revising the paper and addressing my concerns.
> >
> > The new results in the revision look quite positive:
> > 1) The comparison to (Mrowca et al., [1]) indicates that your model has higher performance.  I particularly like the visualization in Figure 2-a highlighting the drawback of the unified dynamics in (Mrowca et al.).
> > 2) The additional generalization results are very nice.
> > 3) Figure 3 addresses the hyperparameter robustness question and unified dynamics question well.
> >
> > I also appreciate the more minor revisions (errorbars, text, ...).
> >
> > Overall I find the paper much more persuasive now, and am changing my review from a 5  (Marginally Below Acceptance Threshold) to an 8 (Top 50% of accepted papers, clear accept).

---

> > > ### Author Response · Authors · 2018-12-03
> > > **Thanks for your comments**
> > >
> > > Dear Reviewer 3,
> > >
> > > That’s great to hear. We’d like to thank you again for your very constructive comments, which have helped us improve the quality of the paper significantly.

---

### Author Response · Authors · 2018-11-20
**General Response**

We want to thank all the reviewers for their insightful comments. Here, we will explain again the contribution of our paper, address some common concerns, and summarize the changes we intend to include in our revisions.

This paper presents a model for learning particle dynamics using a dynamically-built interaction graph. The model works well in simulating and controlling objects under many different states, including rigid bodies, deformable objects, and fluids, both in simulation and on a real robot. We believe that demonstrating the power of learned particle dynamics models on downstream control tasks is an important contribution to the learning and robotics community that should not be undervalued.

We agree with the reviewers that it’s important to compare our paper with other related papers and clarify their differences. Here, we summarize the three key differences between our paper and earlier works that serve as our inspiration [1, 2]:
1) We learn dynamics between particles that are in different physical states (e.g. between rigid bodies and fluids), while other works focused on scenarios where all objects are in the same state, e.g. all soft or rigid bodies (Mrowca et al. [1]), or all fluids (Schenck and Fox [2]).
2) While Mrowca et al. [1] used a static graph, our model uses an interaction graph built dynamically. Since maintaining a dynamic graph is crucial for simulating objects undergoing large deformation like fluids, this modification has enabled our model to work better in the more challenging cases mentioned above.
3) We have applied our model to more challenging control problems, including one on a real-world robot. In the other paper (Mrowca et al. [1]), however, this non-trivial task was not explored at all.

In our revision by Nov. 26, we will include
1) A comparison with Mrowca et al. [1] in all four environments we used, contrasting the two models’ capacity in simulating rigid bodies, elastic deformation, and fluids.
2) Experiments with a unified dynamics model.
3) Systematic analyses of the sensitivity of the hyperparameters.

Please don’t hesitate to let us know if there is any additional comment on the intended changes.

[1] Damian Mrowca, Chengxu Zhuang, Elias Wang, Nick Haber, Li Fei-Fei, Joshua B. Tenenbaum, Daniel L. K. Yamins. "Flexible Neural Representation for Physics Prediction." In NIPS, 2018.
[2] Connor Schenck, Dieter Fox. “SPNets: Differentiable Fluid Dynamics for Deep Neural Networks.” In CoRL 2018.

---

### Author Response · Authors · 2018-11-27
**General Response Cont.**

We thank all reviewers for their comments. We would like to emphasize again that our two main contributions are
1) A particle-based dynamics model that integrates multi-step propagation, hierarchical structure, and dynamic interaction graphs. Our model can simulate objects of different states (rigid bodies, deformable objects, and fluids), significantly outperforming previous methods.
2) Its application to control, with good results on deformable object manipulation both in simulation and on a real robot. In particular, previous papers on learning particle dynamics did not attempt to apply the learned model on control tasks.

We have updated our paper to include the following changes:
1) We have added qualitative and quantitative comparisons with Mrowca et al. [1] on all four environments (Figure 2 and Table 1). Our model significantly outperforms [1], especially on fluid and rigid body simulation. Please see our updated video for a side-by-side comparison.
2) We have included results on generalizing to fluids, deformable objects, and rigid objects that are larger than those in the training set. The results on extrapolation are in Appendix B. Our model generalizes well.
3) We have conducted ablation studies on hyperparameters (Section 4.2). We considered
   a) the propagation step L,
   b) the number of roots in the hierarchy, and
   c) the neighborhood distance d.
4) We have included results on using a unified motion predictor for all objects (Section 4.2).
5) We have also added error bars and confidence intervals for all quantitative metrics.
6) We have updated Figure 1 and the introduction as suggested by reviewers.

[1] Damian Mrowca, Chengxu Zhuang, Elias Wang, Nick Haber, Li Fei-Fei, Joshua B. Tenenbaum, Daniel L. K. Yamins. "Flexible Neural Representation for Physics Prediction." In NIPS, 2018.

---

### Meta-Review · Area_Chair1 · 2018-12-15
**interesting model learning framework**

**Confidence:** 5
**Recommendation:** Accept (Poster)

**Metareview:**

The paper proposes a particle based framework for learning object dynamics. A scene is represented by a hierarchical graph over particles, edges between particles are established dynamically based on Euclidean distance. The model is used for model predictive control, and there is also one experiment with a particle graph built from a real scene as opposed to simulation.

All reviewers agree that the architectural changes over previous relational networks  are worthwhile and merit publication. They also suggest to tone down the ``dynamic” part of the graph construction by stating that edges are determined based on a radius. In particular, previous works also consider similar addition of edges during collisions, quoting Mrowca et al. "Collisions between objects are handled by dynamically defining pairwise collision relations ... between leaf particles..." which suggests that comparison against a baseline for Mrowca et al. that uses a static graph is not entirely fair. The authors are encouraged to repeat the experiment without disabling such dynamic addition of edges.